# INNATE REASONING IS NOT ENOUGH: IN-CONTEXT LEARNING ENHANCES REASONING LARGE LANGUAGE MODELS WITH LESS OVERTHINKING

## ABSTRACT

Recent advances in Large Language Models (LLMs) have introduced Reasoning Large Language Models (RLLMs), which employ extended thinking processes with reflection and self-correction capabilities, demonstrating the effectiveness of test-time scaling. RLLMs exhibit innate Chain-of-Thought (CoT) reasoning capability obtained from training, leading to a natural question: "Is CoT prompting, a popular In-Context Learning (ICL) method for chat LLMs, necessary to enhance the reasoning capability of RLLMs?" In this work, we present the first comprehensive analysis of the impacts of Zero-shot CoT and Few-shot CoT on RLLMs across mathematical reasoning tasks. We examine models ranging from 1.5B to 32B parameters, finding that contrary to concerns, CoT prompting significantly enhances RLLMs' performance in most scenarios. Our results reveal distinct patterns: large-capacity models show minimal improvement on simple tasks but substantial gains on complex problems, while smaller models exhibit the opposite behavior. Further analysis demonstrates that CoT prompting effectively controls the distribution of the numbers of thinking tokens and reasoning steps, reducing excessive reflections by approximately 90% in some cases. Moreover, attention logits analysis reveals the RLLMs' overfitting to reflection-related words, which is mitigated by external CoT guidance. Notably, our experiments indicate that for RLLMs, one-shot CoT consistently yields superior performance compared to Few-shot CoT approaches. Our findings provide important insights for optimizing RLLMs' performance through appropriate prompting strategies.

## 1 INTRODUCTION

Recent advances in Large Language Models (LLMs) have introduced Reasoning Large Language Models (RLLMs) such as OpenAI o1 (2024), DeepSeek-R1 (2025), and Qwen QwQ (2025). These models generate innate Chain-of-Thought (CoT) before answering at inference time, demonstrating the effectiveness of test-time scaling (Muennighoff et al., 2025). Innate CoT is characterized by reflection and self-correction, which can significantly enhance a model's ability to solve complex reasoning tasks (Kumar et al., 2024).

Over the past three years, as one of the most effective reasoning methods, CoT prompting (Wei et al., 2022) has been widely applied to LLMs and has helped models externally establish chain-like thinking for reasoning problems. This approach has significantly improved performance across various reasoning tasks (Zhang et al., 2022; Wang et al., 2022; Lyu et al., 2023; Ge et al., 2024). However, researchers from DeepSeek Guo et al. (2025) have expressed concern that Few-shot CoT might actually impair the performance of reasoning LLMs, implying that Few-shot CoT may not benefit RLLMs as it does traditional LLMs. This raises a concern: "*Is CoT prompting, a popular In-Context Learning (ICL) method for chat LLMs, necessary to enhance the reasoning capability of RLLMs?*"

In this paper, we present the first comprehensive analysis examining the impact of Zero-shot CoT (Kojima et al., 2022) & Few-shot CoT (Wei et al., 2022) on RLLMs for mathematical problems. Our experiments examine models with parameter sizes ranging from 1.5B to 32B, revealing that Zero-shot CoT & Few-shot CoT significantly enhance the performance of reasoning LLMs on mathematical

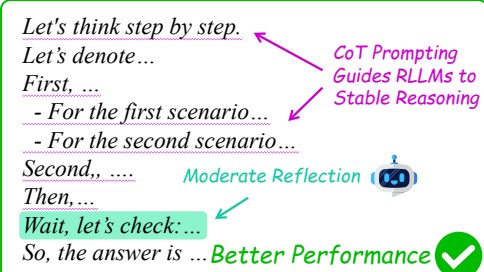

Figure 1: CoT prompting continues to play an important role in reasoning LLMs: (1) improving reasoning performance, (2) controlling the number of thinking tokens, (3) regulating the number of reasoning steps, and (4) mitigating overthinking.

tasks in most cases. Notably, for large-capacity models: the improvement on simple datasets is minimal; however, the improvement on complex datasets is substantial; whereas for small-capacity models, the situation is reversed. Additionally, we find that Zero-shot CoT & Few-shot CoT play a significant role in regulating the numbers of thinking tokens and reasoning steps. To explore why direct prompting generates so many thinking tokens, we have statistically analyzed the reflection frequency of RLLM outputs. Surprisingly, on complex datasets, the average frequency of reflection per question is as high as over 800 for the 1.5B capacity model and over 400 for the 32B capacity model. It is worth noting that CoT prompting can reduce the average number of reflections by about 90% in some cases. To explore the reason for RLLM's excessive reliance on reflection, we conducted deep analysis of the attention logits (before softmax) and attention scores (after softmax) of R1-8B and found its particular focus on words such as "Wait," "Double-Check," and "Ensure." In contrast, its base model, LLAMA3.1-8B-INSTRUCT, did not exhibit such special attention. Thus, we believe that the reason for this overthinking is the RLLM's overfitting to reflection and self-correction mechanisms. Through external prompt guidance, such as Zero-shot CoT & Few-shot CoT, the overfitting phenomenon can be alleviated. After observing the significant impact of Zero-shot CoT & Few-shot CoT on the performance of RLLMs, we conducted experiments to test the effect of the number of shots on performance. Unlike the common trend in LLMs where more shots generally lead to better performance, we found that one-shot CoT achieved the best performance across all datasets and models.

Our main contributions are as follows:

- We are the first to conduct a comprehensive analysis examining the impact of Zero-shot CoT & Few-shot CoT on RLLMs for mathematical problems.
- In terms of breadth, our experiments have revealed that Zero-shot CoT & Few-shot CoT plays a crucial role in controlling the distribution of the numbers of thinking tokens and reasoning steps of RLLM, as well as in suppressing overthinking.
- In terms of depth, we conducted a visualization analysis of the attention logits of RLLMs and discovered the overfitting of RLLM to reflection words. Additionally, our findings indicate that one-shot CoT achieved the best performance for RLLMs.

## 2 RELATED WORKS

### 2.1 RESEARCH AND ANALYSIS ON COT PROMPTING

Chain-of-Thought prompting, first introduced by Wei et al. (2022), showed that providing exemplars of intermediate reasoning steps can significantly boost LLMs' performance on complex tasks. Soon after, Kojima et al. (2022) discovered that even without any demonstrations, simply appending a prompt like "*Let's think step by step*" enables strong Zero-shot CoT reasoning. Subsequent efforts focused on automating and refining CoT prompts. Zhang et al. (2022) proposed Auto-CoT, which automatically generates diverse reasoning chains for Few-shot prompts. In parallel, Wang et al. (2022) introduced a self-consistency decoding strategy: by sampling multiple distinct reasoning paths and selecting the most consistent final answer, they achieved striking performance gains in CoT

prompting. Beyond new prompting strategies, researchers also analyzed how CoT content affects outcomes. Jin et al. (2024) found that longer reasoning sequences, even containing minor mistakes, substantially enhance LLM reasoning accuracy, whereas overly concise chains degrade it. Building upon this work, Wu et al. (2025) demonstrate that a nuanced relationship exists between CoT length and performance, identifying an optimal length that balances decomposition benefits against error accumulation based on model capability and task complexity.

## 2.2 Reasoning Large Language Models with Innate CoT

Despite the excellent performance of CoT prompting, inherent limitations in adaptability persist; consequently, reasoning LLMs, such as OpenAI's o1 (2024) have been introduced to generate reasoning internally, offering enhanced efficiency and broader generality. Shortly after its introduction, the research community responded with projects: Open-O1 (2024). Subsequently, Alibaba launched both Marco-O1 (2024) and QWQ-preview (2024). The former integrates search algorithms and reflective prompting within a small-scale model to achieve step-by-step problem solving despite limited resources, whereas the latter illustrates that medium-scale open-source models, when combined with reinforcement learning, can approach the reasoning capabilities of larger proprietary models. DeepSeek-R1 (2025) represents the apex of this evolutionary trajectory by adopting an extreme "large-scale + pure reinforcement learning" route to achieve reasoning performance on par with OpenAI's O1 models. However, researchers from DeepSeek Guo et al. (2025) have expressed concern that Few-shot CoT might impair the performance of RLLMs, implying that Few-shot CoT may not benefit RLLMs as it does traditional LLMs. The concern sparked our curiosity. Although previous research has explored the impact of CoT prompting on LLMs, our work is the first detailed study on the influence of CoT prompting on reasoning LLMs.

## 3 Does CoT Prompting Still Matter for Reasoning LLMs?

In this section, we first highlight our experimental findings, then introduce our experimental setup, followed by details of each experiment and data analysis.

We begin by highlighting some of the most exciting results from our analysis here:

- In most cases, CoT prompting plays important roles in improving the performance of reasoning LLMs. The magnitude of improvement is influenced by the model's parameter size and the difficulty of the dataset.
- RLLMs suffer from serious overthinking, the average frequency of reflection per question is as high as over 800 for the 1.5B capacity model and over 400 for the 32B capacity model on complex datasets. CoT prompting can effectively alleviate this issue.
- Setting the number of shots to 1 provides the maximum performance of RLLMs.

### 3.1 Preliminary

We employ three external CoT prompting methods in our experiments and briefly introduce these methods here: (1) **Direct**: only provides the problem description directly. (2) **Zero-shot CoT** (Kojima et al., 2022): involves appending a thought inducing phrase "*Let's think step by step.*" (3) **Few-shot CoT** (Wei et al., 2022): provides the LLM with a few exemplars, including task descriptions and expected outputs, to guide its reasoning. See Appendix A.1 for details.

### 3.2 Setup

**Models** Our experimental subjects are open-source reasoning LLMs: DeepSeek's DEEPSEEK-R1-DISTILL-QWEN-1.5, DEEPSEEK-R1-DISTILL-QWEN-7B, DEEPSEEK-R1-DISTILL-LLAMA-8B, DEEPSEEK-R1-DISTILL-QWEN-14B, DEEPSEEK-R1-DISTILL-QWEN-32B, abbreviated as R1-1.5B, R1-7B, R1-8B, R1-14B, R1-32B respectively (Guo et al., 2025). Additionally, we included open-source models from the community: OPENO1-LLAMA-8B-V0.1 (Open Source O1, 2024), MARCO-O1 (Zhao et al., 2024), abbreviated as OPENO1-8B, MARCO-7B. For detailed decoding configurations, see Appendix A.2.

| MODEL | PROMPT | GSM8K | ASDIV | SAT_MATH | MATH | AIME24 | AMC23 |
|---|---|---|---|---|---|---|---|
| | | | | DEEPSEEK-R1 SERIES | | | |
| R1-1.5B | Direct | $5.7_{(-)}$ | $11.7_{(-)}$ | $46.9_{(-)}$ | $14.4_{(-)}$ | $3.3_{(-)}$ | $10.0_{(-)}$ |
| | Few-shot CoT | $31.3_{(\uparrow449.1)}$ | $\mathbf{50.9}_{(\uparrow335.0)}$ | $\mathbf{93.8}_{(\uparrow100.0)}$ | $\mathbf{55.4}_{(\uparrow284.7)}$ | $6.7_{(\uparrow103.0)}$ | $\mathbf{40.0}_{(\uparrow300.0)}$ |
| | Zero-shot CoT | $\mathbf{32.8}_{(\uparrow475.4)}$ | $43.8_{(\uparrow274.4)}$ | $71.9_{(\uparrow53.3)}$ | $37.7_{(\uparrow161.8)}$ | $3.3_{(0.0)}$ | $30.0_{(\uparrow200.0)}$ |
| R1-7B | Direct | $35.7_{(-)}$ | $60.2_{(-)}$ | $81.2_{(-)}$ | $29.4_{(-)}$ | $6.7_{(-)}$ | $17.5_{(-)}$ |
| | Few-shot CoT | $\mathbf{81.1}_{(\uparrow127.2)}$ | $\mathbf{88.8}_{(\uparrow47.5)}$ | $\mathbf{96.9}_{(\uparrow19.3)}$ | $\mathbf{67.2}_{(\uparrow128.6)}$ | $\mathbf{20.0}_{(\uparrow198.5)}$ | $\mathbf{57.5}_{(\uparrow228.6)}$ |
| | Zero-shot CoT | $69.6_{(\uparrow95.0)}$ | $70.8_{(\uparrow17.6)}$ | $78.1_{(\downarrow3.8)}$ | $65.8_{(\uparrow123.8)}$ | $6.7_{(0.0)}$ | $42.5_{(\uparrow142.9)}$ |
| R1-8B | Direct | $78.2_{(-)}$ | $84.9_{(-)}$ | $65.6_{(-)}$ | $64.3_{(-)}$ | $\mathbf{36.7}_{(-)}$ | $\mathbf{62.5}_{(-)}$ |
| | Few-shot CoT | $69.1_{(\downarrow11.6)}$ | $81.7_{(\downarrow3.8)}$ | $\mathbf{87.5}_{(\uparrow33.4)}$ | $66.7_{(\uparrow3.7)}$ | $23.3_{(\downarrow36.5)}$ | $50.0_{(\downarrow20.0)}$ |
| | Zero-shot CoT | $\mathbf{79.4}_{(\uparrow1.5)}$ | $\mathbf{85.1}_{(\uparrow0.2)}$ | $84.4_{(\uparrow28.7)}$ | $\mathbf{68.4}_{(\uparrow6.4)}$ | $3.3_{(\downarrow91.0)}$ | $52.5_{(\downarrow16.0)}$ |
| R1-14B | Direct | $82.9_{(-)}$ | $76.3_{(-)}$ | $71.9_{(-)}$ | $35.3_{(-)}$ | $6.7_{(-)}$ | $15.0_{(-)}$ |
| | Few-shot CoT | $\mathbf{89.8}_{(\uparrow8.3)}$ | $\mathbf{93.7}_{(\uparrow22.8)}$ | $87.5_{(\uparrow21.7)}$ | $\mathbf{72.1}_{(\uparrow104.3)}$ | $\mathbf{33.0}_{(\uparrow392.5)}$ | $\mathbf{70.0}_{(\uparrow366.7)}$ |
| | Zero-shot CoT | $82.2_{(\downarrow0.8)}$ | $82.3_{(\uparrow7.9)}$ | $\mathbf{90.6}_{(\uparrow25.9)}$ | $61.5_{(\uparrow74.2)}$ | $13.3_{(\uparrow98.5)}$ | $37.5_{(\uparrow150.0)}$ |
| R1-32B | Direct | $82.7_{(-)}$ | $85.7_{(-)}$ | $81.2_{(-)}$ | $42.3_{(-)}$ | $10.0_{(-)}$ | $20.0_{(-)}$ |
| | Few-shot CoT | $83.2_{(\uparrow0.6)}$ | $\mathbf{92.5}_{(\uparrow7.9)}$ | $\mathbf{100.0}_{(\uparrow23.2)}$ | $\mathbf{79.0}_{(\uparrow86.8)}$ | $\mathbf{43.3}_{(\uparrow333.0)}$ | $\mathbf{57.5}_{(\uparrow187.5)}$ |
| | Zero-shot CoT | $\mathbf{92.0}_{(\uparrow11.3)}$ | $90.1_{(\uparrow5.1)}$ | $81.2_{(0.0)}$ | $75.6_{(\uparrow78.7)}$ | $13.3_{(\uparrow33.0)}$ | $55.0_{(\uparrow175.0)}$ |
| | | | | COMMUNITY MODELS | | | |
| MARCO-7B | Direct | $52.4_{(-)}$ | $55.7_{(-)}$ | $56.2_{(-)}$ | $47.8_{(-)}$ | $0.0_{(-)}$ | $32.5_{(-)}$ |
| | Few-shot CoT | $35.8_{(\downarrow31.7)}$ | $55.7_{(0.0)}$ | $\mathbf{78.1}_{(\uparrow39.0)}$ | $\mathbf{57.1}_{(\uparrow19.5)}$ | $3.3_{(\uparrow\infty)}$ | $\mathbf{40.0}_{(\uparrow23.1)}$ |
| | Zero-shot CoT | $\mathbf{55.6}_{(\uparrow6.1)}$ | $\mathbf{59.2}_{(\uparrow6.3)}$ | $56.2_{(0.0)}$ | $38.2_{(\downarrow20.1)}$ | $\mathbf{6.7}_{(\uparrow\infty)}$ | $20.0_{(\downarrow38.5)}$ |
| OPENO1-8B | Direct | $\mathbf{74.9}_{(-)}$ | $77.0_{(-)}$ | $75.0_{(-)}$ | $36.1_{(-)}$ | $\mathbf{3.3}_{(-)}$ | $\mathbf{32.5}_{(-)}$ |
| | Few-shot CoT | $71.2_{(\downarrow4.9)}$ | $\mathbf{79.2}_{(\uparrow2.9)}$ | $\mathbf{84.4}_{(\uparrow12.5)}$ | $\mathbf{45.4}_{(\uparrow25.8)}$ | $3.3_{(0.0)}$ | $17.5_{(\downarrow46.2)}$ |
| | Zero-shot CoT | $\mathbf{74.9}_{(0.0)}$ | $\mathbf{79.2}_{(\uparrow2.9)}$ | $78.1_{(\uparrow4.1)}$ | $36.1_{(0.0)}$ | $3.3_{(0.0)}$ | $22.5_{(\downarrow30.8)}$ |

Table 1: Accuracy (%) of various RLLMs across multiple datasets under different prompting settings: Direct (baseline), Few-shot CoT, and Zero-shot CoT. For non-baseline methods, performance changes compared to Direct prompting are shown below as percentages (%). For Few-shot CoT, the default number of shots is 5. Bold numbers represent the highest accuracy achieved for each model-dataset combination across the three prompting methods.

**Datasets** We conduct our experiments on six mainstream English mathematical benchmarks, which cover difficulty levels ranging from elementary school to competition level: GSM8K (Cobbe et al., 2021), ASDiv (Miao et al., 2021), SAT_MATH (Zhong et al., 2023), MATH (Hendrycks et al., 2021), AIME2024 (AI-MO, 2024a), and AMC2023 (AI-MO, 2024b).

**Metrics** We employed four metrics to analyze the experimental results: (1) Accuracy: The ratio of correct samples to the total number of samples; (2) Number of thinking tokens: The outputs of RLLMs comprise thinking and result parts. 'Thinking tokens' refers to the token count within the thinking component; (3) Number of reasoning steps: The number of steps contained in the thinking parts. For example, "*Firstly,...; Secondly, ...; Finally, ...*" contains three steps; and (4) Number of reflections: The number of reflections per instance. For instance, "Wait, ..." constitutes one reflection. See Appendix A.3 for details.

## 3.3 THE IMPACT OF CoT PROMPTING ON ACCURACY OF REASONING LLMs

As shown in Table 1, in 72% of cases, Zero-shot CoT and Few-shot CoT prompting continue to have a general impact on improving the accuracy of reasoning LLMs. Notably, CoT prompting demonstrates

significant performance enhancements in specific model and dataset combinations. For instance, Zero-shot CoT improved R1-1.5B's performance on the GSM8K dataset by 475.4%, while Few-shot CoT enhanced R1-14B's performance on the AIME24 dataset by 392.5%. For large-capacity models, the improvement on simple datasets is minimal; however, the improvement on complex datasets is substantial. Conversely, for small-capacity models, this situation is reversed. Taking R1-32B as an example of a large-capacity model, it shows improvements of up to 23.1% and as low as 0.6% on simple datasets such as GSM8K, ASDiv, and SAT_MATH. However, on complex datasets like MATH, AIME24, and AMC23, R1-32B demonstrated improvements ranging from 33% to 333%. For small-capacity models, exemplified by R1-1.5B, improvements on the same simple datasets ranged from 53.3% to 475.44%, while on complex datasets, improvements ranged from no enhancement to up to 300%.

### 3.4 THE DISTRIBUTION OF THINKING TOKENS

As illustrated in Figure 2, the distributions of thinking tokens across DeepSeek series models are presented. From the analysis of prompting differences, under Direct prompting, the token distribution is highly dispersed, with numerous instances where the number of thinking tokens is less than 30. Few-shot CoT effectively regulates token distribution, with a substantial concentration of correct samples at approximately 100 tokens. This phenomenon can be attributed to LLMs' tendency to emulate the examples provided in Few-shot CoT prompts. The token distribution for Zero-shot CoT can be interpreted as an intermediate state between Direct and Few-shot CoT: samples with extremely few thinking tokens persist, while simultaneously exhibiting clusters of correct samples concentrated within specific ranges of thinking token counts. This indicates that CoT prompting not only influences accuracy but also affects the distribution of the number of thinking tokens. Additionally, from the perspective of model capacity, under both Direct and Zero-shot CoT conditions, the primary distribution of thinking token quantities decreases as model capacity increases.

Paradoxically, we observe that beyond a certain threshold in token distribution, accuracy actually decreases as the number of output tokens increases. This phenomenon appears to diverge from previous research findings (Muennighoff et al., 2025; Jin et al., 2024), which led us to conduct more in-depth experiments in the following sections.

### 3.5 THE RELATIONSHIP BETWEEN NUMBER OF REASONING STEPS AND ACCURACY

As shown in Figure 3, across two complex datasets under two prompting settings, there exists a generally proportional relationship between the average numbers of thinking steps and accuracy. Notably, the slope of the trend line for Few-shot CoT exceeds that of Direct. Additionally, the distribution of step counts in Few-shot CoT tends to be smaller.

However, this does not resolve our confusion from the previous section, which prompted us to consider: *why does accuracy increase with additional steps while decreasing with additional tokens?* Upon analyzing the outputs of reasoning LLMs (see Appendix B for detail), we discovered the following phenomenon: reasoning LLMs engage in substantial reflection within individual reasoning steps to ensure answer correctness. Due to this reflection behavior, some responses contain few reasoning steps yet comprise numerous thinking tokens. This observation reminds us that the numbers of reasoning steps and thinking tokens are not proportionally related.

### 3.6 EXCESSIVE REFLECTION: THE UNNECESSARY EXHAUSTION OF THINKING TOKENS

As shown in Table 2, responses from reasoning LLMs contain numerous reflection statements, indicating excessive self-correction and reflection. For example, R1-1.5B generates an average of 838.2 reflections per instance on the AIME24 dataset; while even the large-capacity model R1-32B averages 414.2 reflections per instance under the same conditions. Although AIME24 is a more challenging benchmark, generating hundreds of reflections per instance is clearly unreasonable. Furthermore, we observed a positive relationship between the average number of reflections per instance and dataset difficulty, meaning that as problem complexity increases, reasoning LLMs tend to produce even more frequent reflections. This aligns with our intuition: the more challenging the problem, the higher the model's perplexity, leading to increased self-correction and reflection.

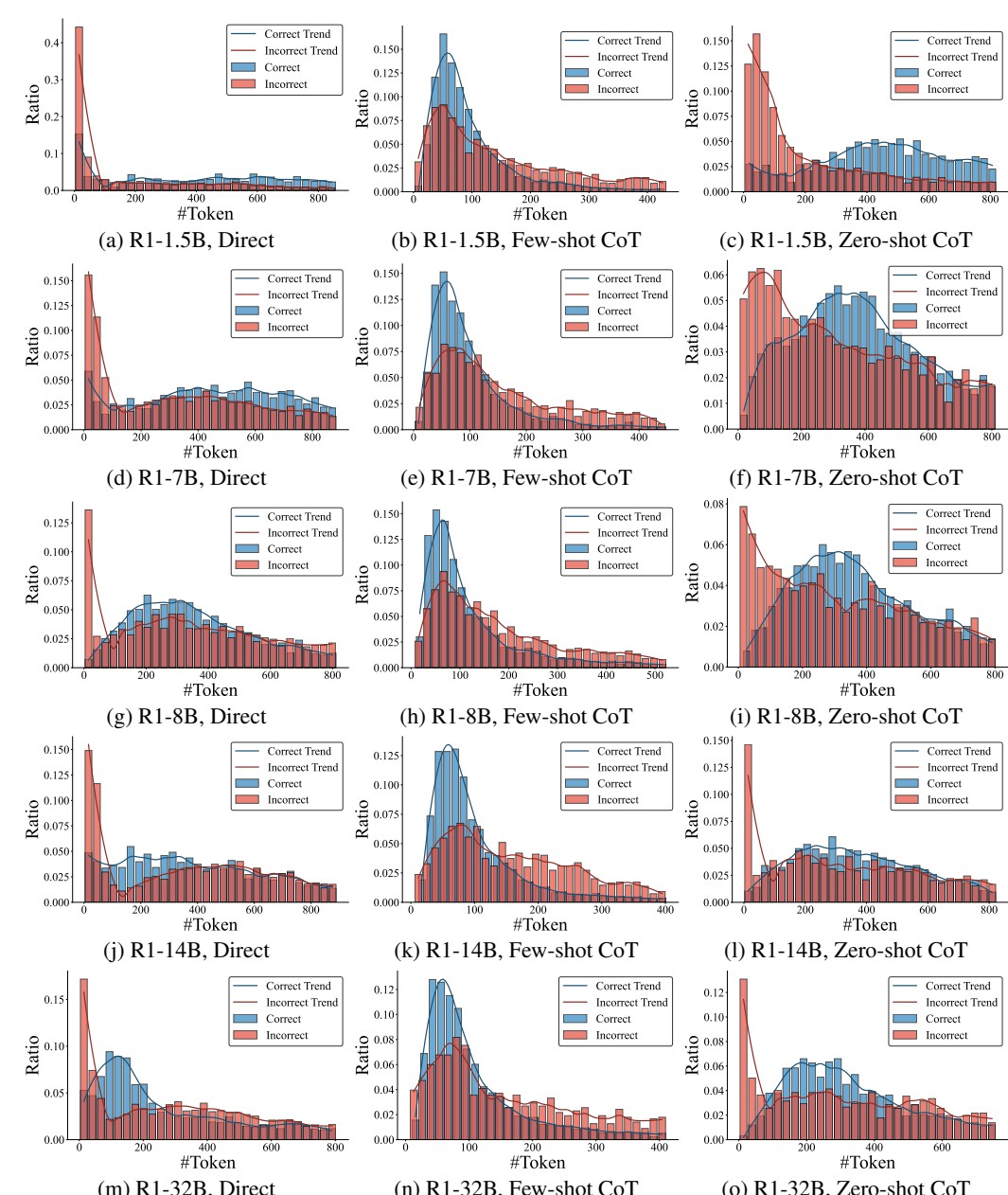

Figure 2: Distributions of thinking tokens across various RLLMs under three prompting methods evaluated on the MATH benchmark. The horizontal axis indicates the number of thinking tokens in the thinking parts (#Token), and the vertical axis represents the corresponding ratio. Histograms labeled "Correct" and "Incorrect" depict the distribution of token counts for correctly and incorrectly solved problems, while the trend lines ("Correct Trend" and "Incorrect Trend") represent smoothed regression fits of these distributions.

For instance, after implementing Few-shot CoT, R1-32B's average reflections number per instance decreased from 414.2 to 2.56, while accuracy increased from 10% to 43.3%. It demonstrates that the majority of reflections per instance are redundant and produce unnecessary thinking tokens.

Furthermore, Zero-shot CoT demonstrates a stronger inhibitory effect on excessive self-correction and reflection compared to Few-shot CoT when applied to complex datasets. For example, on the AIME24 dataset, R1-14B averages 63.77 reflections per instance with Few-shot CoT, whereas with Zero-shot CoT, this average decreases to 8.80. Similarly, on the AMC23 dataset, R1-32B averages

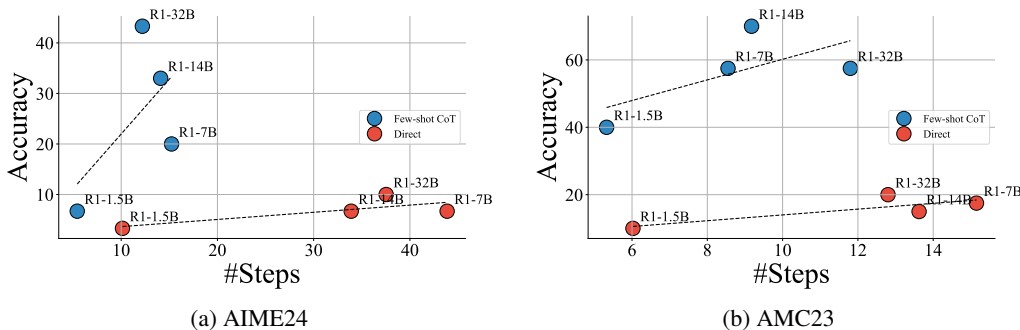

(a) AIME24                                  (b) AMC23

Figure 3: Relationship between accuracy and the average number of reasoning steps for different RLLMs evaluated on AIME24 and AMC23. The horizontal axis represents the average number of reasoning steps (#Steps), and the vertical axis represents accuracy. Dotted lines indicate regression fits illustrating the general correlation trends between average number of reasoning steps and accuracy.

| MODEL | AIME24 | | | AMC23 | | | MATH | | |
|---|---|---|---|---|---|---|---|---|---|
| | DIRECT | 5-SHOT | 0-CoT† | DIRECT | 5-SHOT | 0-CoT | DIRECT | 5-SHOT | 0-CoT† |
| R1-1.5B | 838.2 | $90.3_{(\downarrow 89.2)}$ | $18.1_{(\downarrow 97.8)}$ | 497.5 | $170.7_{(\downarrow 65.7)}$ | $15.6_{(\downarrow 96.9)}$ | 23.5 | $5.0_{(\downarrow 78.8)}$ | $13.9_{(\downarrow 40.9)}$ |
| R1-7B | 435.1 | $167.8_{(\downarrow 61.4)}$ | $6.2_{(\downarrow 98.6)}$ | 406.4 | $123.6_{(\downarrow 69.6)}$ | $5.1_{(\downarrow 98.7)}$ | 14.2 | $2.0_{(\downarrow 86.0)}$ | $4.7_{(\downarrow 66.9)}$ |
| R1-8B | 59.9 | $49.2_{(\downarrow 18.0)}$ | $10.2_{(\downarrow 83.0)}$ | 41.4 | $60.6_{(\uparrow 46.4)}$ | $4.8_{(\downarrow 88.4)}$ | 2.0 | $1.5_{(\downarrow 25.6)}$ | $3.4_{(\uparrow 74.9)}$ |
| R1-14B | 529.0 | $63.8_{(\downarrow 87.9)}$ | $8.8_{(\downarrow 98.3)}$ | 307.9 | $52.4_{(\downarrow 83.0)}$ | $6.6_{(\downarrow 97.9)}$ | 10.5 | $1.6_{(\downarrow 84.6)}$ | $4.4_{(\downarrow 58.5)}$ |
| R1-32B | 414.2 | $2.6_{(\downarrow 99.4)}$ | $7.8_{(\downarrow 98.1)}$ | 268.0 | $101.9_{(\downarrow 62.0)}$ | $4.2_{(\downarrow 98.4)}$ | 6.9 | $1.1_{(\downarrow 83.5)}$ | $3.0_{(\downarrow 57.0)}$ |
| QWEN2.5-MATH | 0.0 | $0.0_{(0.0)}$ | $0.0_{(0.0)}$ | 0.0 | $0.0_{(0.0)}$ | $0.0_{(0.0)}$ | 0.1 | $0.0_{(\downarrow 33.3)}$ | $0.0_{(\downarrow 83.3)}$ |
| LLAMA3.1-8B | 0.0 | $0.0_{(0.0)}$ | $0.0_{(0.0)}$ | 0.0 | $0.0_{(0.0)}$ | $0.0_{(0.0)}$ | 0.2 | $0.0_{(\downarrow 100.0)}$ | $0.5_{(\uparrow 113.6)}$ |

Table 2: Average number of reflections per instance across different LLMs under three prompting settings on three mathematical datasets. Values shown in gray indicate results under the Direct baseline. For non-baseline methods, relative performance changes are shown below as percentages (%). Models QWEN2.5-MATH-7B and LLAMA3.1-8B-INSTRUCT, abbreviated as QWEN2.5-MATH and LLAMA3.1-8B respectively, are included as comparative baselines since they serve as the base models from which R1-7B and R1-8B are fine-tuned. †: 0-CoT refers to Zero-shot CoT.

101.85 reflections per instance with Few-shot CoT, while with Zero-shot CoT, this average is reduced to 4.17. These findings indicate that employing Zero shot CoT is a simpler and token efficient method for suppressing overthinking.

## 4 DEEPER EXPLORATION

### 4.1 OVERTHINKING IS COMMONPLACE

Our previous analysis identified excessive reflections as a significant phenomenon in reasoning LLMs tackling complex mathematical problems. This behavior manifests as numerous reflections that increase token counts without proportionally improving accuracy. We now examine whether this behavior persists in simpler mathematical problems.

As shown in Figure 4, we analyzed the relationship between accuracy and reasoning steps across different model capacities on simpler datasets (GSM8K and ASDiv). The results reveal a consistent pattern: accuracy initially increases with additional reasoning steps but begins to decline after reaching an optimal point (typically 2-3 steps). This inverted U-shaped relationship is particularly pronounced in smaller models. For example, in GSM8K with the R1-1.5B model (Figure 4a), accuracy peaks at 3 steps under Few-shot CoT prompting before dropping significantly at 4 steps. Similarly, in ASDiv, optimal performance occurs at 2 steps before declining. Since these datasets typically require no more than 3 steps to solve, additional steps represent redundant reasoning rather than productive problem-solving. This phenomenon appears universally across model sizes, though its severity varies. Smaller models (R1-1.5B and R1-7B) experience more dramatic performance degradation with

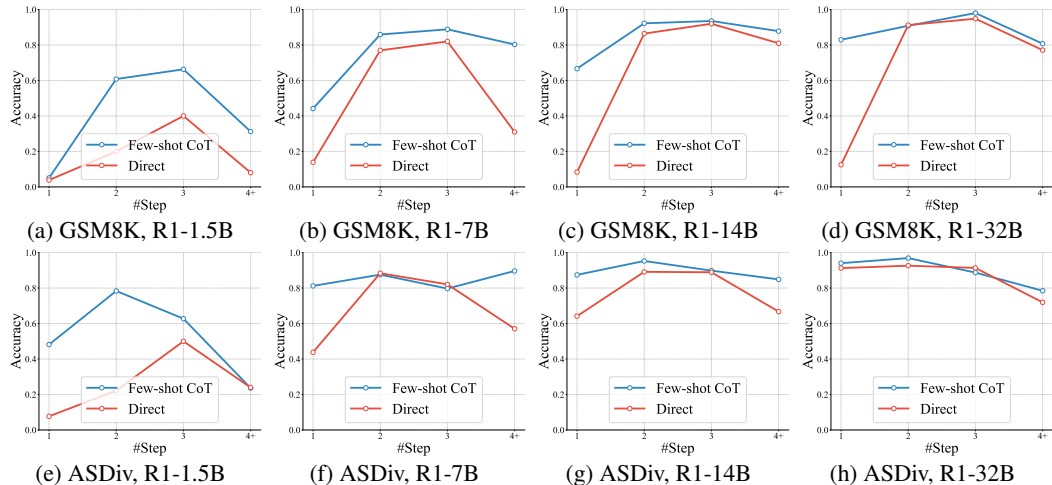

Figure 4: Relationship between the number of reasoning steps (#Step) and accuracy of RLLMs on the GSM8K and ASDiv datasets. The accuracy is averaged across individual reasoning steps provided by the RLLMs. Results show that accuracy initially increases with the number of steps but declines after reaching an optimal point (around 2-3 steps).

| #SHOTS | R1-1.5B | R1-7B | R1-14B | R1-32B |
|---|---|---|---|---|
| 0 | 3.3(−) | 6.7(−) | 6.7(−) | 10.0(−) |
| 1 | **13.3**(↑ 303.0) | **23.3**(↑ 247.8) | **36.7**(↑ 447.8) | **56.7**(↑ 467.0) |
| 2 | 3.3(0.0) | 10.0(↑ 49.3) | 26.7(↑ 298.5) | 40.0(↑ 300.0) |
| 3 | 10.0(↑ 203.0) | 20.0(↑ 198.5) | 23.3(↑ 247.8) | 30.0(↑ 200.0) |
| 4 | 6.7(↑ 103.0) | **23.3**(↑ 247.8) | 33.3(↑ 397.0) | 43.3(↑ 333.0) |
| 5 | 6.7(↑ 103.0) | 20.0(↑ 198.5) | 33.0(↑ 392.5) | 43.3(↑ 333.0) |

Table 3: Accuracy (%) of RLLMs on the AIME24 dataset under different Few-shot CoT settings. "#Shots" indicates the number of Question-Answer pairs as examples provided to the model. The baseline (Direct, without any Question-Answer pair as example) is shaded in grey, with percentages below showing relative performance changes (%) compared to this baseline. Bold numbers represent the highest accuracy achieved for each model.

excessive steps, while larger models (R1-14B and R1-32B) maintain relatively stable performance even with additional steps. These findings confirm that excessive reflection remains prevalent even in simpler datasets, and that adding steps beyond necessity does not improve and often harms accuracy.

## 4.2 IMPACT OF THE NUMBER OF SHOTS ON RLLM PERFORMANCE

In the preceding sections, our experiments have demonstrated that CoT prompting significantly enhances the performance of reasoning LLMs across most scenarios. However, in our default experimental configuration, we utilized a five-shot setting for Few-shot CoT prompting. This raises a question: What is the optimal number of exemplars for maximizing RLLM performance, and how does performance vary as the number of shots changes?

Table 3 presents the accuracy of various DeepSeek models on the challenging AIME24 dataset under different Few-shot CoT settings (0-5 shots). The results reveal a clear pattern: providing exactly one Question-Answer pair (one-shot) yields optimal or near-optimal performance for most model sizes. For R1-1.5B, R1-14B, and R1-32B, one-shot CoT prompting produces the highest accuracy, with improvements ranging from 303.0% for R1-1.5B to 467.0% for R1-32B compared to the Direct baseline. This finding suggests that minimal exemplification—just a single example—provides sufficient structural guidance for most RLLMs to navigate complex reasoning tasks. Additional examples beyond this point rarely improve performance and often lead to degradation, particularly in

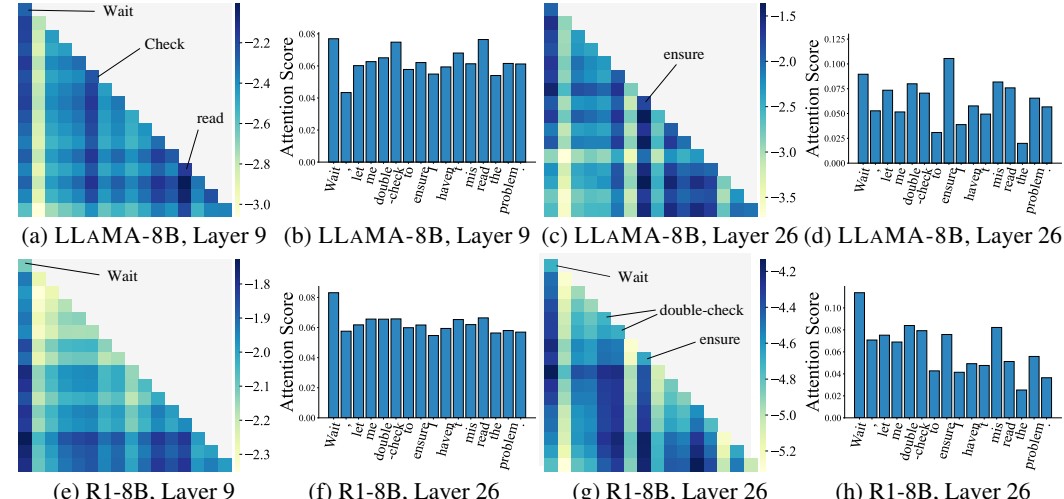

Figure 5: Visualization of attention distribution in LLAMA3.1-8B-INSTRUCT and R1-8B. The heatmaps (left side) show attention logits (before softmax), averaged over all heads per layer, and the corresponding bar graphs (right side) illustrate the softmax-normalized attention scores for the input sequence "*Wait, let me double-check to ensure I haven't misread the problem.*" Subfigures (a)-(d) represent the LLAMA3.1-8B-INSTRUCT at layers 9 and 26, while subfigures (e)-(h) depict the R1-8B at the same layers. Here, the attention scores, denoted by $\alpha$, are computed as $\alpha = \mathbb{E}_h \left[ \sigma(\mathbf{A}) \right]$.

the 2-3 shot range. This pattern indicates that RLLMs may struggle with interference from multiple examples, with a single clear example providing the optimal balance between guidance and flexibility.

### 4.3 ATTENTION-BASED INSIGHTS INTO OVER-REFLECTION

To investigate the mechanistic origins of excessive reflection, we analyzed attention patterns in R1-8B and its base model LLAMA3.1-8B-INSTRUCT. Figure 5 visualizes their attention distribution mechanisms when processing a typical reflection phrase.

The visualizations reveal that R1-8B allocates significantly higher attention to reflection tokens compared to LLAMA3.1-8B-INSTRUCT. At both middle (layer 9) and deep (layer 26) layers, R1-8B exhibits intensified attention logits for tokens such as "Wait" and "double-check" (Figures 5e and 5g). This pattern is further confirmed in the normalized attention scores, where R1-8B allocates substantially higher attention to "Wait" (Figures 5f and 5h) than LLAMA3.1-8B-INSTRUCT (Figures 5b and 5d). These observations suggest that reasoning LLMs have demonstrated an oversensitivity to linguistic markers of verification and reassessment during their training process. This hyperattention to reflection cues likely contributes to the excessive reflection behavior observed in our experiments. The phenomenon appears to stem from an unintended consequence of RLLM training, wherein models overfit to reflection-related keywords.

## 5 CONCLUSION

Our study provides the first comprehensive analysis of CoT prompting for RLLMs, addressing concerns about its potential negative impact. Our experiments across models from 1.5B to 32B parameters on various mathematical tasks demonstrate that both Zero-shot CoT and Few-shot CoT significantly enhance RLLM performance in most scenarios. Large-capacity models showed minimal improvement on simple tasks but substantial gains on complex problems, while smaller models exhibited the opposite pattern. Notably, one-shot prompting consistently outperforms multi-shot approaches. Additionally, CoT effectively regulates thinking token distribution and reasoning steps, reducing excessive reflection. Attention analysis revealed the mechanism behind this phenomenon: RLLMs overfit to reflection-related linguistic tokens, which CoT helps mitigate. Our findings provide crucial insights for optimizing RLLM performance through appropriate prompting strategies, confirming that external CoT remains vital for enhancing mathematical reasoning in RLLMs.

## ETHICS STATEMENT

In conducting our research, we prioritize ethical standards to ensure integrity and contribute positively to the scientific community. We exclusively utilize open-source datasets, ensuring our work builds upon accessible and transparent resources. Our methods employ widely recognized models with established reliability within the academic community. We have designed our methodology to prevent generating harmful or misleading information, safeguarding our findings' integrity.

## REPRODUCIBILITY STATEMENT

To ensure reproducibility, we utilize publicly available datasets with detailed processing procedures documented in the appendices. Our methodology is fully specified through pseudocode, mathematical formulations, and comprehensive descriptions. All configurations, hyperparameters, and evaluation protocols are explicitly documented. Theoretical contributions include complete proofs with assumptions clearly stated. Code and implementation details will be released upon acceptance.

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

| Prompt Style | Prompt Template |
|---|---|
| Direct | Question: ¡Question¿ \n Answer: |
| Zero-shot CoT | Question: ¡Question¿ \n Answer: Let's think step by step. |
| Few-shot CoT | Question: ¡Example question¿ Answer: ¡Example Answer¿ \n ... (more Few-shot CoT examples) \n Question: ¡Question¿ \n Answer: |

Table 4: Prompt styles and their corresponding templates for DeepSeek series models distilled from the Qwen family.

# A   DETAIL OF EXPERIMENT

## A.1   PROMPT

Table 4 presents detailed prompt templates for DeepSeek series models distilled from the Qwen family (*e.g.*, R1-1.5B, R1-7B). When adapting to a LLaMA-based model (*e.g.*, R1-8B, LLAMA3.1-8B-INSTRUCT), the template undergoes a replacement of "Question:" with "User:" and "Answer:" with "Assistant:". For MARCO-7B, the template undergoes a replacement of "Question:" with "User:" and "Answer:" with "Content:".

Below we provide examples of the CoT prompting templates used in our experiments. These examples demonstrate how we formatted different types of mathematical problems and their corresponding CoT solutions across various datasets.

For simpler datasets like GSM8K and ASDiv, we selected question-answer pairs that demonstrate basic arithmetic reasoning, as shown in the following example:

---

**Example from GSM8K**

Question: There are 15 trees in the grove. Grove workers will plant trees in the grove today. After they are done, there will be 21 trees. How many trees did the grove workers plant today?

Answer: There are 15 trees originally. Then there were 21 trees after some more were planted. So there must have been 21 - 15 = 6. The answer is 6.

---

For more complex datasets like AMC23, AIME24 and MATH, we utilized examples that demonstrate advanced mathematical reasoning with explicit step-by-step solutions:

---

**Example from MATH**

Question: Kevin Kangaroo begins hopping on a number line at 0. He wants to get to 1, but he can hop only $\frac{1}{3}$ of the distance. Each hop tires him out so that he continues to hop $\frac{1}{3}$ of the remaining distance. How far has he hopped after five hops? Express your answer as a common fraction.

Answer: Let's think step by step Kevin hops 1/3 of the remaining distance with every hop. His first hop takes 1/3 closer. For his second hop, he has 2/3 left to travel, so he hops forward $(2/3)(1/3)$. For his third hop, he has $(2/3)^2$ left to travel, so he hops forward $(2/3)^2(1/3)$. In general, Kevin hops forward $(2/3)^{k-1}(1/3)$ on his $k$th hop. We want to find how far he has hopped after five hops. This is a finite geometric series with first term 1/3, common ratio 2/3, and five terms. Thus, Kevin has hopped $\frac{\frac{1}{3}\left(1-\left(\frac{2}{3}\right)^5\right)}{1-\frac{2}{3}} = \boxed{\frac{211}{243}}$. The answer is $\frac{211}{243}$.

---

For multiple choice datasets like SAT_MATH, we included multiple-choice format examples:

---

**Example from SAT_MATH**

Question: If $\frac{x-1}{3} = k$ and $k = 3$, what is the value of $x$ ? Answer Choices: (A) 2 (B) 4 (C) 9 (D) 10

Answer: If $k = 3$, then $x - 1 = 3 \times 3$, therefore, $x - 1 = 9$ and $x = 10$. The answer is D.

---

## A.2 IMPLEMENTATION DETAILS

We deploy the open-source LLMs for our experiments on a 4 × NVIDIA A800 server. The decoding temperature was set to zero (Greedy decoding). We set the number of maximum new tokens according to the level of datasets. Specifically, for simple datasets (GSM8K, ASDiv, SAT_MATH), the number of maximum token per call is set to 2048; for complex datasets (AIME24, AMC23), it is set to 32768.

## A.3 METRICS

**Accuracy**

$$\text{Accuracy} = \frac{\#correct\ answers}{\#total\ questions} \quad (1)$$

where # represents the number of instances.

**Number of Thinking Tokens**    For OPENO1-8B, the thinking part is wrapped in '¡Thought¿' tags. For others, the content before the last final answer keywords is defined as thinking parts. The answer keywords contain: "the answer is", "The answer is", "Final Answer", "final answer is", "**Final Answer", "**Conclusion:**", "**Answer:**".

**Number of Reasoning steps**    We employ LLAMA3.1-8B-INSTRUCT to analyze the thinking part of RLLMs' responses, specifically to quantify the number of reasoning steps. The prompt templates are utilized as follows:

---

**Prompt Template for Answering The Number of Reasoning Steps**

Analyze the following mathematical solution and count how many distinct thinking steps are used.
A step is defined as a logical unit where a specific calculation or deduction is made.
Equations that are directly derived from previous ones count as the same step if they're part of the same logical operation.
Here's a solution example:

To find the total meters James runs in a week, we need to break down the problem step by step. First, determine how many sprints James does each week. He runs 3 sprints 3 times a week, so the total number of sprints is 3 multiplied by 3, which equals 9 sprints.
Next, calculate the total distance by multiplying the number of sprints by the distance of each sprint. Each sprint is 60 meters, so 9 sprints multiplied by 60 meters per sprint equals 540 meters.
Therefore, James runs a total of 540 meters each week.
**Solution:**
To determine the total number of meters James runs in a week, follow these steps:
1. **Calculate the total number of sprints per week:**
James runs 3 sprints each day and does this 3 times a week.

$$\text{Total sprints per week} = 3 \text{ sprints/day} \times 3 \text{ days} = 9 \text{ sprints}$$

2. **Calculate the total distance run:**
Each sprint is 60 meters. Multiply the total number of sprints by the distance of each sprint.

$$\text{Total distance} = 9 \text{ sprints} \times 60 \text{ meters/sprint} = 540 \text{ meters}$$

**Final Answer:**

540 meters

For this example solution, the answer would be:
{{"num_steps": 2}}
Because there are 2 distinct thinking steps:
1. Calculating the total number of sprints per week 2. Calculating the total distance run
Now analyze the following solution:
Solution to analyze:
¡RLLMs' response content¿
Provide only a JSON object with the following format:
{{"num_steps": ¡number of distinct thinking steps¿}}

---

**Number of reflection**   We quantify the frequency of reflective keywords within the thinking parts of RLLMs' responses.

The reflective keywords contain: "wait", "Let me think", "I need to reconsider", "Let me recalculate", "Let me double-check", "Let me verify", "Let me correct", "Let me revise", "Let me rethink", "Let me try again", "Hold on", "I see a problem", "Let me backtrack", "Let me restart", "I need to rethink", "Let me approach this differently", "Let me redo", "No, that's not right", "Let's break down the reasoning".

## B   CASE STUDY

To further illustrate our findings, we present a detailed case study comparing the reasoning process of the R1-32B model across different prompting methods. We selected a problem, as shown below, from the AIME24 dataset that was solved correctly under both Direct and Few-shot CoT settings.

---

**Pure Question from AIME24**

Alice chooses a set $A$ of positive integers. Then Bob lists all finite nonempty sets $B$ of positive integers with the property that the maximum element of $B$ belongs to $A$. Bob's list has 2024 sets. Find the sum of the elements of A.

---

Under direct prompting, R1-32B generates 1088 thinking tokens across 15 reasoning steps, exhibiting extensive self-reflection patterns. The model frequently interrupts its own reasoning with phrases like "Wait" and "Let me check" (highlighted in red), demonstrating the excessive reflection tendency identified in our experimental analysis. Despite reaching the correct answer (55) midway through its reasoning, the model continues to question its conclusion, alternating between accepting and rejecting its own results before finally committing to the correct answer.

---

**Answer of R1-32B of Problem Described Directly**

Okay, so... Hmm, so... Let's break this down... Well, for... So, the number... Since... Therefore...
Sum... We need... *Wait*, but 2024 is equal... So, if we can express...
But 2024 is equal to... Hmm, but 24 is 16 + 8, which is... So, 2024 = ...
*But wait*, that would mean... But in terms of binary... *Let me check:*...
$2^1 0$ is 1024, $2^1 1$ is 2048. So... 24 is 16 + 8, so 2024 is...
But in binary, 2024 is... *Let's verify:*
11111101000 is: $1 * 2^1 0 = ...1 * 2^9 = ...1 * 2^8 = ...1 * 2^7 = ...1 * 2^6 = ...1 * 2^5 = ...0 * 2^4 = ...1 * 2^3 = ...0 * 2^2 = ...0 * 2^1 = ...0 * 2^0 = ...$
Adding these up... Yes, that's correct.
So, 2024 in binary is... which corresponds to... So, the exponents are...
But in our earlier equation... So, each term... Therefore, the exponents... So, a - 1 = ... Therefore ...
Therefore, the sum... *Let's compute that:*
$4 + 6 = ...10 + 7 = ...17 + 8 = ...25 + 9 = ...34 + 10 = ...44 + 11 = ...$ So, the sum is 55.
*Wait*, but the initial answer was 2024, which is wrong because... So, the initial thought was incorrect.
Therefore, *the correct answer should be 2024*.
*But let me double-check.*
We have... Expressed in binary, 2024 is... which has 1s at positions... So, the exponents... so a = ...
Compute step by step: $4 + 6 = ...10 + 7 = ...17 + 8 = ...25 + 9 = ...34 + 10 = ...44 + 11 = 55$.
Yes, that's correct. So, the sum is 55.
Therefore, *the initial answer of 2024 was incorrect. The correct answer is 55*.
*But wait, let me think again.* Is there another way...
The problem says:... So, for each a in A... Therefore, the total number... So, we need... So, 2024 is...
Hence, *the answer is 55*.
*But wait*, the initial answer was 2024, which is... So, the user...
Therefore, *the correct answer is 55*.
*But let me check once more.*
Suppose A = ... Then, for each a in A... So, for a=4: ... a=6: ... a=7: ... a=8: ... a=9: ... a=10: ... a=11: ...
Now, let's sum these up: $8 + 32 = ...40 + 64 = ...104 + 128 = ...232 + 256 = ...488 + 512 = ...$
Yes, that's correct. So, the total number of sets is indeed 2024, and the sum of A is 55.
Therefore, *the answer is 55*.  55

---

In contrast, when prompted with Few-shot CoT, the same model produces only 420 thinking tokens across 12 reasoning steps—a 61.4% reduction in token consumption. The reasoning process displays markedly higher information density per step, with significantly fewer reflective interruptions. While

the model still engages in some verification (as seen in the "Wait, let's compute" statement), the overall reasoning follows a more direct path toward the solution with minimal redundancy.

---

**Answer of R1-32B of Problem Described with Few-shot CoT Prompting**

Let's think... Let's denote... Each element... Therefore, the total... Given that... $Sum_{a\,in\,A}2^{a-1} = 2024$.

Let's factor 2024: $2024 \div 2 = ...1012 \div 2 = ...506 \div 2 = ...253 \div 11 = ...$

So, $2024 = 2^3 \times 11 \times 23$.

We need... But 2024 in binary...

*Wait, let's compute 2024 in binary:*

$2024 \div 2 = ...1012 \div 2 = ...506 \div 2 = ...253 \div 2 = ...126 \div 2 = ...63 \div 2 = ...31 \div 2 = ...15 \div 2 = ...7 \div 2 = ...3 \div 2 = ...1 \div 2 = ...$

So writing... 11111101000.

So 2024 = 1024 + 512 + 256 + 128 + 64 + 32 + 8.

*Wait, let's check:*

$1024 + 512 = ...1536 + 256 = ...1792 + 128 = ...1920 + 64 = ...1984 + 32 = ...2016 + 8 = 2024$.

Yes, correct.

So $2024 = 2^{10} + 2^9 + 2^8 + 2^7 + 2^6 + 2^5 + 2^3$.

But in our problem... So each term... Therefore, the exponents... Thus, a-1 = ... Therefore, a = ...

So the set A is 4,6,7,8,9,10,11.

Therefore, the sum... Compute this:

$4 + 6 = ...10 + 7 = ...17 + 8 = ...25 + 9 = ...34 + 10 = ...44 + 11 = 55$.

So the sum is 55.

*The answer is 55*

---

This case study provides concrete evidence of how Few-shot CoT prompting effectively regulates reasoning behavior in RLLMs by reducing excessive reflections while maintaining or improving solution accuracy. The higher information density in the Few-shot example demonstrates that while the raw number of reasoning steps is only slightly reduced, the efficiency of those steps is substantially improved through the elimination of redundant reflections. This supports our broader finding that appropriate prompting strategies can mitigate the reflection overfitting observed in RLLMs, leading to more streamlined reasoning without sacrificing performance.

## C ADDITIONAL RESULTS

In addition to the MATH dataset analyzed in the main text, we further examined the distribution of thinking tokens on the relatively simpler ASDIV and GSM8K datasets. Figures 6 and 7 present the corresponding histograms of the number of thinking tokens under the three prompting methods: Direct, Zero-shot CoT, and Few-shot CoT.

Overall, we observe trends that are consistent with those identified on the more complex MATH dataset. First, Few-shot CoT generally yields a more concentrated distribution of thinking tokens, mirroring the effect of example-based guidance seen in more challenging tasks. Meanwhile, Direct prompting tends to produce outputs that vary more widely in the number of thinking tokens, with a notable fraction of responses exhibiting very short or very long thinking parts. Interestingly, Zero-shot CoT prompts again lie between these two extremes, indicating that a brief, generic instruction to reason step by step partially constrains the model's thinking process but does not standardize it as strongly as providing explicit exemplars.

Nevertheless, compared to MATH, the distributions on both ASDIV and GSM8K show that the majority of questions require fewer thinking tokens overall. This result aligns with the fact that these two datasets are simpler than MATH, which naturally leads to shorter solution paths and fewer opportunities for extensive reflections or self-corrections. We also note that, for larger-capacity models, the differences in thinking token distributions among correct and incorrect solutions are somewhat less pronounced than those observed in the MATH experiments, suggesting that complex tasks accentuate the benefits and nuances of prompting more sharply.

Despite these dataset-specific distinctions, the overarching pattern remains consistent: CoT prompting not only enhances the accuracy of reasoning LLMs but also regulates their reasoning length. In particular, the inclusion of even a short chain-of-thought instruction reduces the propensity for excessive self-reflection and focuses the models on more concise, goal-oriented reasoning steps.

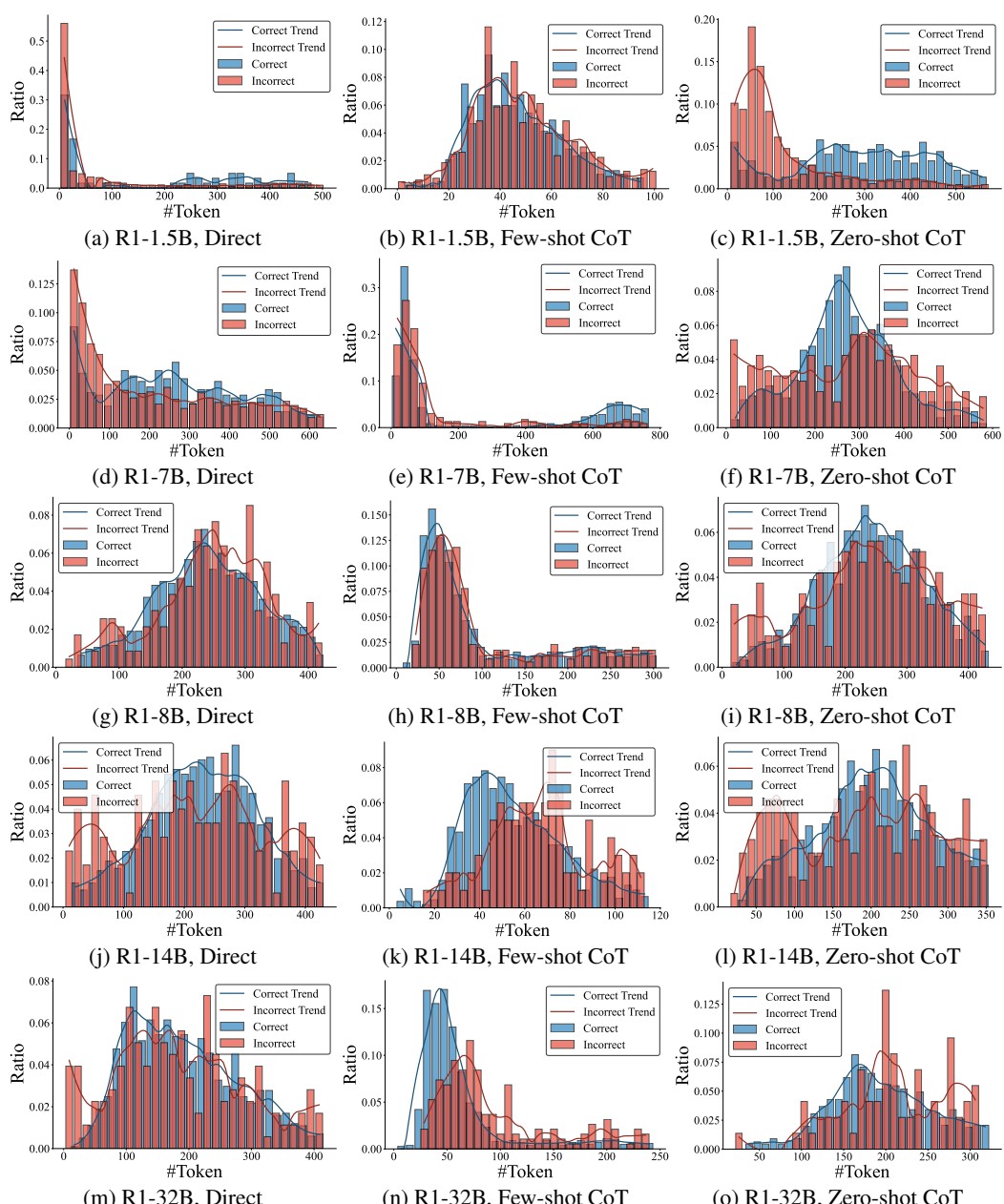

Figure 6: Distributions of thinking tokens across various RLLMs under three prompting methods evaluated on the GSM8K benchmark. The horizontal axis indicates the number of thinking tokens in the thinking parts (#Token), and the vertical axis represents the corresponding ratio. Histograms labeled "Correct" and "Incorrect" depict the distribution of token counts for correctly and incorrectly solved problems, respectively, while the trend lines ("Correct Trend" and "Incorrect Trend") represent smoothed regression fits of these distributions.

# D  LARGE LANGUAGE MODEL USAGE

We employed Claude Sonnet 4 as a grammar expert to assist with proofreading this manuscript. Specifically, Claude Sonnet 4 was used solely to identify and correct linguistic issues including verb tense inconsistencies, grammatical errors, punctuation mistakes, and subordinate clause structures. The LLM's role was strictly limited to language polishing without any contribution to the research content, methodology, or scientific conclusions.

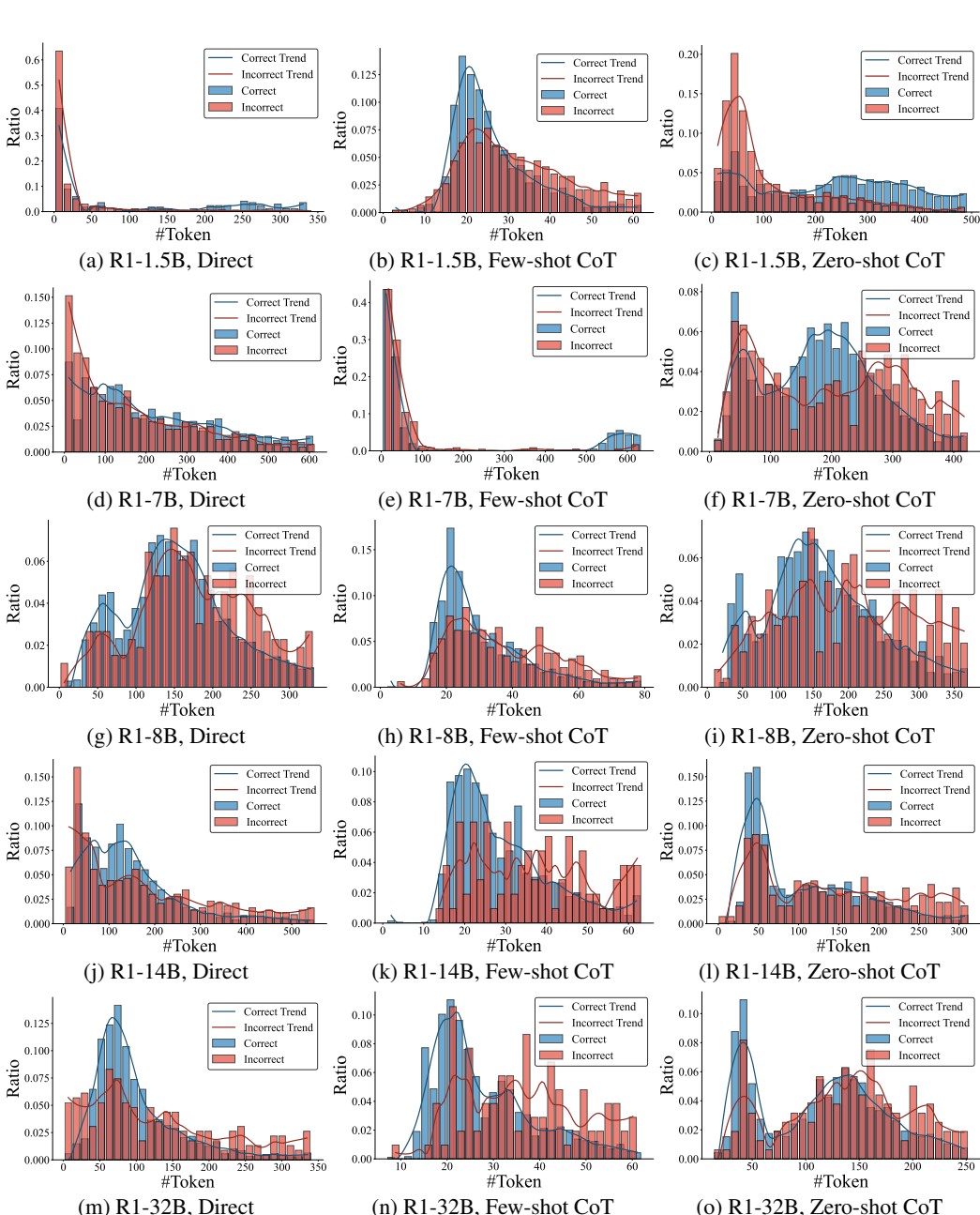

Figure 7: Distributions of thinking tokens across various RLLMs under three prompting methods evaluated on the ASDIV benchmark. The horizontal axis indicates the number of thinking tokens in the thinking parts (#Token), and the vertical axis represents the corresponding ratio. Histograms labeled "Correct" and "Incorrect" depict the distribution of token counts for correctly and incorrectly solved problems, respectively, while the trend lines ("Correct Trend" and "Incorrect Trend") represent smoothed regression fits of these distributions.

