# OpenReview forum: "Innate Reasoning is Not Enough: In-Context Learning Enhances Reasoning Large Language Models with Less Overthinking"
_ICLR.cc/2026/Conference — ICLR 2026 Conference Withdrawn Submission_

### Official Review · Reviewer_jnpY · 2025-10-31

**Soundness:** 3
**Presentation:** 2
**Contribution:** 3
**Rating:** 4
**Confidence:** 3

**Summary:**

The paper examines the effects of Few-shot CoT prompting methods on the performance of Reasoning Large Language Models (RLLMs) across five mathematical tasks. It investigates deepseek series models ranging from 1.5B to 32B parameters and reveals that Few-shot prompting improves RLLM performance on complex tasks, reduces overthinking, and one-shot CoT outperforms Few-shot CoT, with attention analysis revealing overfitting to reflection-related words.

**Strengths:**

1. The author provides an in-depth discussion on solving mathematical problems using the DeepSeek R1-like reasoning model. The paper’s figures and tables are clear and easy to understand, effectively presenting the author's findings in a visually intuitive manner.

2. I believe that a detailed analysis of the internal reasoning process of reasoning models is essential. The attention maps, length statistics, step counts, and frequency of reflection-related terms provided by the authors offer valuable insights, providing a deeper perspective on the reasoning process.

**Weaknesses:**

1. The experiments are conducted exclusively on mathematical datasets, which might limit the generalizability of the findings to other reasoning tasks.

2.  The analytical experiments (except Table 1) are only conducted on the DeepSeek-series models, which may raise concerns about the representativeness of the findings. I’m wondering whether similar issues, such as overthinking, are present in other reasoning models like OpenAI's o1 or Gemini to confirm the broader applicability of the results.

3.  Previous studies [1, 2, 3] have suggested that overthinking in reasoning models can negatively affect performance. The approach of using CoT prompting without example reasoning to reduce the length of the reasoning process is a natural and intuitive solution to mitigate overthinking.

[1] Hassid, Michael et al. “Don't Overthink it. Preferring Shorter Thinking Chains for Improved LLM Reasoning.” ArXiv abs/2505.17813 (2025): n. pag.

[2] Cuadron, Alejandro et al. “The Danger of Overthinking: Examining the Reasoning-Action Dilemma in Agentic Tasks.” ArXiv abs/2502.08235 (2025): n. pag.

[3] Liu, Ryan et al. “Mind Your Step (by Step): Chain-of-Thought can Reduce Performance on Tasks where Thinking Makes Humans Worse.” ArXiv abs/2410.21333 (2024): n. pag.

**Questions:**

Please see the Weaknesses.

---

### Official Review · Reviewer_fNF3 · 2025-10-31

**Soundness:** 2
**Presentation:** 3
**Contribution:** 2
**Rating:** 4
**Confidence:** 4

**Summary:**

This paper studies whether explicit Chain-of-Thought (CoT) prompting still helps “reasoning LLMs” (RLLMs) that already produce latent thoughts (e.g., DeepSeek-R1 variants, OpenO1). The authors claim: (i) Zero-shot and Few-shot CoT generally improve accuracy across six math benchmarks; (ii) CoT reduces “overthinking” by curbing thinking-token length, reasoning steps, and reflection frequency; (iii) “one-shot” CoT tends to outperform multi-shot; and (iv) attention visualizations suggest RLLMs over-attend to reflection tokens like “Wait” / “double-check.”

**Strengths:**

1. Timely question & clear takeaway. Despite innate test-time thinking, external CoT often helps—sometimes dramatically—on complex math datasets (e.g., large improvements reported on AIME/AMC).

2. Consistent “one-shot” observation. Table 3 shows one-shot Few-shot CoT often peaks on AIME24, with much smaller or negative gains beyond one shot.

3. Behavior analyses. Distributions of thinking tokens/steps and reflection counts are informative; attention maps offer a plausible mechanism for “over-reflection.”

**Weaknesses:**

1. Novelty feels incremental. Empirically comparing CoT styles on RLLMs is useful, but methodologically this is largely a descriptive study (prompting + measurement) rather than a new algorithm or causal intervention. The “first comprehensive analysis” claim is strong given the evaluation is only on the math benchmarks; no other reasoning tasks are included; the contribution is primarily systematic measurement on math reasoning tasks.

2. While Table 3 shows strong one-shot results, it’s unclear how general this is across alternative demonstrations, prompt orderings, or instruction styles. The provided templates are single-instance illustrations; more systematic prompt ablations are needed.

3. Scope limited to English math. Claims of general RLLM behavior would be stronger with non-math reasoning (code, science QA, planning) and multilingual tasks.

**Questions:**

Do the one-shot gains persist under different exemplars, orders, and styles? Any variance analysis?

Can you report confidence intervals or run multiple seeds (with sampling) to assess effect stability? Current results are largely point estimates with greedy decoding.

---

### Official Review · Reviewer_cpKM · 2025-11-01

**Soundness:** 2
**Presentation:** 3
**Contribution:** 2
**Rating:** 2
**Confidence:** 4

**Summary:**

This paper investigates the effects of chain-of-thought (CoT) prompting on the performance of reasoning large language models (RLLMs). The paper experiments with both zero-shot and few-shot in-context learning (ICL) and compares to direct prompting (no CoT), on a wide range of open-source models on six English-language benchmarks (GSM8K, ASDiv, SAT_MATH, MATH, AIME2024, AMC2023). Models are evaluated w.r.t. their accuracies on each task as well as the number of thinking tokens generated, the number of reasoning steps provided within their thinking, and the number of reflections.

The authors find that large-capacity models exhibit substantial improvements with CoT prompting on complex tasks and less so on simpler ones. The opposite can be observed for small-capacity models. The authors also show that few-shot CoT prompting regulates the thinking token distribution. They also identify a proportional relationship between the number of reasoning steps and task accuracy and observe over-reflection behaviour for RLLMs.

**Strengths:**

This paper provides a detailed and extensive analysis of the reasoning behaviour of a wide range of open-source RLLMs in the context of existing English-language reasoning benchmarks. The presented results shed light on which prompting methods (direct, zero-shot CoT, few-shot CoT) are effective and efficient for such RLLMs.

**Weaknesses:**

* Technical contributions: The technical contributions of this paper are incremental and do not contain a substantial amount of novelty. The paper uses existing methods to prompt LLMs for existing benchmarks and analyses their behaviour. This is interesting and exciting to see but the scope would need to be widened for this to be considered for publication.
* Methodology: The used methodology around the computation of reasoning steps and reflections in this paper requires additional validation. In particular, the computation of reasoning steps is conducted with another LLM (LLAMA3.1-8B-INSTRUCT) and the authors do not report on methods to verify and confirm the correctness of using an LLM in this context (in other words, how confident can we be that the LLM correctly counts the number of reasoning steps for other LLMs). Likewise, the number of reflections is computed based on a keyword list mentioned in the appendix (containing phrases such as “Let me think” and ”Let me redo”). The authors do not mention experiments assessing the accuracy of that method w.r.t. a set of golden-labelled reasoning traces.

**Questions:**

Could you provide more details on the methods used to compute reasoning steps and reflections? Were any steps involved to assess the correctness of such methods?

---

### Official Review · Reviewer_NvXN · 2025-11-04

**Soundness:** 2
**Presentation:** 2
**Contribution:** 2
**Rating:** 2
**Confidence:** 4

**Summary:**

The paper revisits the effect of {direct, zero-shot CoT*, one-shot CoT, few-shot CoT} prompting on reasoning LLMs such as DeepSeek R1. The authors study the task accuracy and other metrics such as number of tokens, reasoning steps (occurrence of keywords like wait, double-check) when applying these various prompting schemes. The authors consider the DeepSeek-R1-Distill-Qwen series of models, as well as community models OpenO1-Llama-8B-v0.1 and Marco-o1. The authors consider various math tasks: GSM8K, ASDiv, SAT_MATH, MATH, AIME24, AMC23. These are the main findings from the paper:

- I1. CoT prompting significantly enhances accuracy in many scenarios.
- I2. CoT prompting affects reasoning length and Few-shot CoT notably creates a concentrated distribution
- I3. Few-shot-CoT results in less reasoning steps, which leads to decreased accuracy.
- I4. The number of reflections in reasoning models are significantly higher than that of base models.
- I5. There is an inversed U-shape relationship between number of reasoning steps and accuracy, suggesting the existence of overthinking.
- I6. The optimal number of examples in few-shot prompting was typically 1.
- I7. Reasoning models allocate significant attention scores to reflection tokens like wait and double-check.

*Zero-shot CoT refers to including "let's think step-by-step" in the prompt.

**Strengths:**

- The paper re-investigates the effect of established prompting schemes that had previously been widely studied on legacy LLMs, but not explicitly studied on reasoning LLMs like DeepSeek-R1, to the best of my knowledge.
- The authors consider a wide range of model sizes and math reasoning tasks.
- The presentation is clear.

**Weaknesses:**

- There is limited evidence to support the generality of the findings. The following points could be improved:
  - W1. Only one set of few-shot examples were used. The effect of few-shot prompting can hugely differ based on the specific examples used. E.g., longer examples may induce longer reasoning, contradicting the claimed findings of the paper. This is a **major concern**.
  - W2. Investigation on the behavior of reasoning models is mainly focused on a single family of reasoning models: DeekSeep-R1-Distill-Qwen.
  - W3. The investigation is limited to math tasks.
  - W4. The results do not consistently support the claims in some cases, even with the limited scope of investigation. For example, while Table 1 shows that *either* zero-shot or few-shot CoT generally improves performance over direct prompting, it is less generally the case that *both* zero-shot and few-shot CoT improves performance. There are many cases where zero-shot CoT performs better than direct prompting but few-shot CoT performs worse, and vice-versa.
- W5. The findings offer low actionable utility..

**Questions:**

Suggestions:
- The number in Table 1 could be better visually aligned to make it easier to compare values.

---

### Note · Authors · 2026-01-01

I have read and agree with the venue's withdrawal policy on behalf of myself and my co-authors.